# Geochemical Assessment of the Evolution of Groundwater under the Impact of Seawater Intrusion in the Mannar District of Sri Lanka

Samadhi Athauda [1,2,3,†], Yunwen Wang [1,†], Zhineng Hao [1,2,*], Suresh Indika [1,2,3], Isuru Yapabandara [1,2,3], Sujithra K. Weragoda [3,4], Jingfu Liu [5] and Yuansong Wei [1,2,3,*]

1    Research Center for Eco-Environmental Sciences, Chinese Academy of Sciences, Beijing 100085, China; athaudasamadhi@yahoo.com (S.A.); yunwwang@163.com (Y.W.); indika_st@rcees.ac.cn (S.I.); yapaban@gmail.com (I.Y.)
2    University of Chinese Academy of Sciences, Beijing 100049, China
3    China-Sri Lanka Joint Research and Demonstration Center for Water Technology, Ministry of Water Supply, Meewathura, Peradeniya 20400, Sri Lanka; skwera7@gmail.com
4    National Water Supply and Drainage Board, Katugastota 20800, Sri Lanka
5    Institute of Environment and Health, Jianghan University, Wuhan 430056, China; jfliu@rcees.ac.cn
*    Correspondence: znhao@rcees.ac.cn (Z.H.); yswei@rcees.ac.cn (Y.W.)
†    These authors contributed equally to this work.

**Abstract:** Groundwater is an important drinking water resource in the coastal regions of island countries and has suffered from heavy seawater intrusion. However, the areas specifically affected by seawater intrusion and their groundwater hydrogeochemical compositions and evolution processes remain unclear. This study analyzed the hydrogeochemical compositions, water quality, and evolution processes of groundwater in the Mannar district, Sri Lanka, during the dry season. A total of 56 samples were collected from shallow wells and tube wells across the region, and about 64.28% of groundwater samples had good quality (WQI < 100). Geochemical compositions and water quality parameters had a high level in the north and south mainland regions, where they severely suffered from seawater intrusion with a high content of $Cl^-$ and $Na^+$. The geochemical compositions of groundwater in the Mannar district were predominantly affected by rock weathering and/or evaporation processes. Cl-Na and $HCO_3$-Ca facies were the main hydrochemical types, and the corresponding ions were mainly from silicate and halite dissolution. The reverse cation exchange process mainly occurred in seawater intrusion areas. The study highlights the impacts of seawater intrusion on the hydrogeochemical compositions and evolution processes in Mannar region groundwater, which will enhance the understanding of the local water quality and seawater intrusion situation and aid in protecting drinking water safety by routinely monitoring the groundwater quality and implementing targeted desalination techniques in the key areas.

**Keywords:** groundwater; hydrogeochemical composition; water quality; spatial–temporal mapping; seawater intrusion; evolution process

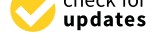



## 1. Introduction

Groundwater serves as a vital freshwater source for domestic, drinking, and agricultural purposes in numerous regions worldwide. However, the quality and accessibility of groundwater reserves are being heavily impacted by both natural factors and anthropogenic activities [1]. Natural factors, such as climate variations [2], mineral dissolution and precipitation, exchange reactions, seasonal rainfall distribution asymmetry, chemical weathering in various rock formations [3], and interactions between soil/rock and groundwater during flow and recharge [4], as well as seawater intrusion [5], play a crucial role in shaping groundwater quality. Conversely, anthropogenic activities include urbanization, agricultural practices [6], industrial pollution, groundwater overextraction [7], and waste

disposal [8]. Seawater intrusion is considered the most important factor that affects the hydrogeochemistry and quality of groundwater in coastal areas [9], which also often face severe freshwater shortages. Seawater intrusion affects the groundwater in coastal regions mainly through rock–water interactions [10], freshwater mixing [11], and geochemical reactions [12], leading to elevated levels of dissolved ions in coastal aquifer freshwater and a decrease in freshwater availability [13]. Seawater intrusion is anticipated to aggravate in some cases, like overgroundwater extraction [14], tectonic activity [15], global warming [16], and sea level rise [17].

Besides the direct influence on freshwater compositions, seawater intrusion significantly affects the natural evolution processes of groundwater in coastal aquifers and furthers the groundwater geochemistry. The interaction between seawater and freshwater alters aquifer geochemical conditions [18], mineral composition [19], seawater infiltration rate [19], and water residence time [20], as well as the consequent diverse chemical reactions. For example, the elevated salinity in groundwater led to a change in ionic strength and redox conditions due to the dissolution of gypsum and halite and the subsequent release of sulfate and chloride ions [19], resulting in increased concentrations of specific chemical species in groundwater [21]. Therefore, it is essential to reveal the complex processes and mechanisms of groundwater evolution in coastal areas suffering from seawater intrusion.

Seawater intrusion and its influence on groundwater evolution are particularly serious in tropical equatorial regions, where most rain falls in a few months each year and coastal aquifers are rich in highly porous sedimentary rocks [22]. The high temperature in these regions leads to intense evapotranspiration and impacts the overall groundwater balance in the aquifer, which is more serious during prolonged dry spells [23]. The highly porous sedimentary rocks favor the flow of seawater into the aquifer with high groundwater recharge rates [24,25]. The sea level rise caused by tidal forces and global warming further aggravates the seawater intrusion. In the South Asian region, a 0.1-m sea level rise promoted seawater intrusion with a decrease in freshwater aquifer thickness from 25 to 10 m [26,27]. Unfortunately, the groundwater geochemistry and quality impacted by seawater intrusion are still insufficiently investigated in developing countries in tropical equatorial regions, greatly hindering the understanding of groundwater in coastal regions and its effect on human health.

Sri Lanka, located in the Indian Ocean, is a tropical-equatorial country and a water-related vulnerability hotspot [28]. The country is particularly susceptible to seawater intrusion due to its coastal geography, sea level fluctuation, and equatorial climates [20,24,29,30]. The northern and northwestern regions of Sri Lanka feature permeable coastal aquifers that are composed of a Miocene limestone belt and the overlain unconsolidated deposits of sand and sandy clays [31]. During the dry season, the region experiences minimal rainfall and semi-arid conditions, leading to reduced groundwater recharge and severe water scarcity [32,33]. These features accelerate the seawater intrusion and its influence on groundwater quality. Therefore, this study investigates the geochemical characteristics and evolution processes of groundwater in the Mannar district, located in the northwestern region of Sri Lanka. To this end, shallow and tube well groundwater samples were collected in the middle of the dry season (May 2022), when intense seawater intrusion is anticipated to occur. Their water quality parameters and the contents of major compositions were analyzed. The groundwater evolution processes and primary impact factors were then identified. This work will provide insights for the evolution process of groundwater in tropical equatorial regions and basis support for groundwater quality management.

## 2. Experimental Section

### 2.1. Study Area and Data Collection

The Mannar district ($8°52'$ N, $80°4'$ E) is situated on the northwestern coast of Sri Lanka (Figure 1). It spans an approximate area of 1996 km$^2$ and is characterized as one of the driest regions in Sri Lanka, with an average annual rainfall of 975 mm/year and evapotranspiration of 2135 mm/year, falling within a semi-arid climate zone. The rainfall,

together with several other recharge components, balances the evaporation loss, including groundwater inflow, surface runoff, recharge from underground flow connections, contributions from coastal limestone aquifers, and water from potential recharge sources like the Giant Tank and Malwathu Oya [34]. The temperature in the region ranges from 23 °C to 35 °C. The population in the district is comparatively low, and the main livelihood activities are agriculture, fishing, and animal husbandry. These activities, plus the high evaporation rates, are the major environmental stressors on the groundwater resources in the area. The Mannar district consists of both an island part and a mainland area (Figure 1). The underlying geological formation primarily consists of Miocene limestone and Quaternary deposits (Figure S1a) [35]. The limestone is composed of sedimentary limestone and sandstone formations. The sedimentary limestone is highly faulted, causing the aquifer to be separated into a series of isolated blocks and groundwater basins. The limestone beds are overlain by unconsolidated Quaternary deposits (Figure S1a) [35], including red earth, yellow–brown sands, dune and beach sands, and lagoonal deposits. The island's geology comprises a Miocene limestone basement and the overlain Holocene wind-blown dune sand (Figure S1b) [29,36].

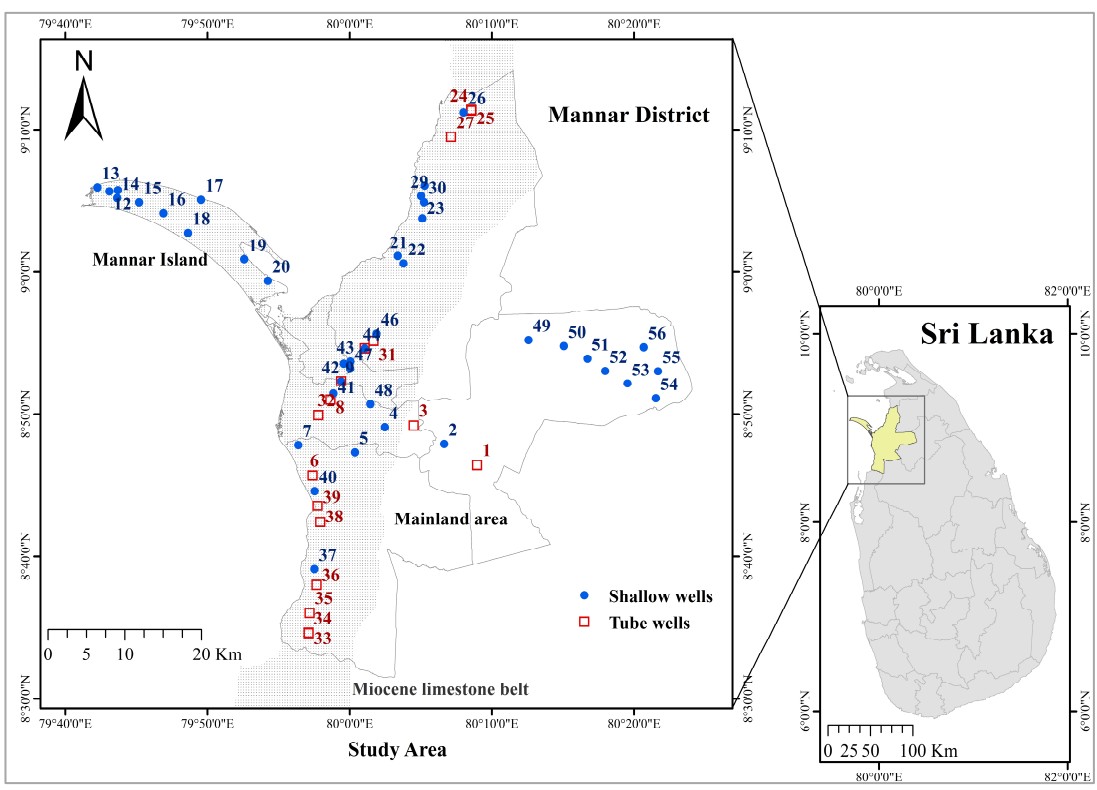

**Figure 1.** Study area and sampling points, including 39 shallow wells (blue points) and 17 tube wells (red points).

*2.2. Sample Collections and Measurements*

Sampling was conducted in the dry season (May 2022), and a total of 56 groundwater samples (17 tube wells and 39 shallow wells) were collected in the Mannar district (Figure 1). A total of 2 L of groundwater was collected and transferred into polypropylene bottles, which were pre-rinsed with 10% (*v/v*) hydrochloric acid and then with deionized water. Physicochemical parameters such as pH, temperature, electrical conductivity (EC), total dissolved solids (TDS), and dissolved oxygen (DO) were measured in the field by using portable test kits (Eutech 01X099414, Thermo Scientific Eutech 01X099414 Inc., Waltham, MA, USA).

The physicochemical analyses of water samples were carried out at the China–Sri Lanka Joint Research and Demonstration Center for Water Technology, Ministry of Water

Supply, Peradeniya. The water samples were filtered with a 0.45 μm filter membrane before conducting geochemical characterization. Inorganic cations ($Ca^{2+}$, $Mg^{2+}$, $Na^+$, and $K^+$) were measured using inductively coupled plasma optical emission spectrometry (PerkinElmer, Optima 8300, Waltham, MA, USA). Inorganic anions ($F^-$, $Cl^-$, $Br^-$, $NO_3^-$, and $SO_4^{2-}$) were measured using ion chromatography (Eco IC, Metrohm, Herisau, Switzerland) equipped with an ion chromatogram pack column (ion exchange resin) and a conductivity detector. The total alkalinity was determined by $H_2SO_4$ (0.01 N) titration with methyl orange as the indicator. Dissolved organic carbon (DOC) was analyzed using a TOC analyzer (MULTI N/C 2100, Analytik-jena, Jena, Germany).

### 2.3. Water Quality Index (WQI) and Seawater Mixing Index (SMI) Calculations

WQI is a numeric indicator employed to indicate the overall water quality in a given area [37]. Here, WQI was calculated by referencing Hameed et al. (2010) [38], which incorporated twelve parameters with assigned weights. These parameters include pH, EC, DO, TDS, alkalinity, $Na^+$, $Ca^{2+}$, $Mg^{2+}$, $F^-$, $Cl^-$, $SO_4^{2-}$, and $NO_3^-$. The calculation of the WQI first needed to normalize the weights of each parameter ($w_i$) (Table S2) based on Equation (1) to obtain the relative weight ($W_i$):

$$W_i = \frac{w_i}{\sum_i^n w_i} \tag{1}$$

Then, twelve parameters ($C_i$) were standardized based on the threshold value set that was provided by the World Health Organization [39] ($S_i$), and the partial WQI score ($Q_i$) of each parameter was calculated as Equation (2). WQI was finally obtained according to Equation (3).

$$Q_i = \frac{C_i}{S_i} \times 100 \tag{2}$$

$$WQI = \sum_{i=1}^{m} W_i Q_i \tag{3}$$

The calculated WQI was categorized into five levels: excellent water (WQI: <50), good water (WQI: 50–100), poor water (WQI: 100–200), very poor water (WQI: 200–300), and unsuitable water (WQI: >300).

SMI, first introduced by Park et al. (2005), is a valuable index for analyzing the mixing potential of groundwater and seawater [24,40,41]. SMI is determined based on the concentrations of four key ions in seawater, i.e., $Na^+$, $Mg^{2+}$, $Cl^-$, and $SO_4^{2-}$. The SMI calculation is shown in Equation (4).

$$SMI = a \times \frac{C_{Na}}{T_{Na}} + b \times \frac{C_{Mg}}{T_{Mg}} + c \times \frac{C_{Cl}}{T_{Cl}} + d \times \frac{C_{SO_4}}{T_{SO_4}} \tag{4}$$

where a, b, c, and d represent the relative fraction of $Na^+$, $Mg^{2+}$, $Cl^-$, and $SO_4^{2-}$ in seawater with values of 0.31, 0.04, 0.57, and 0.08 [40], C represents the concentration of the key ions in terms of mg/L, and T represents the regional threshold values for specific ions and is determined by cumulative probability curves (Figure S2) [42]. SMI above 1 indicates the presence of seawater intrusion [42].

### 2.4. Data Analysis

Statistical analysis plots were generated using Grapher 20 software, and spatial interpolation maps for groundwater parameters were drawn using Arc GIS (version 10.5) software with the inverse distance-weighted interpolation technique [43].

### 2.5. Evolution Mechanisms of Groundwater

Three diagrams were used to reveal the evolution processes and mechanisms of groundwater in Mannar district, i.e., a Gibbs diagram [44], a Piper diagram [45], and a hydrochemical facies evolution diagram (HFE-D) [46]. The Gibbs diagram is useful for

understanding the sources that control groundwater chemistry. It consists of three main regions: rock interaction, evaporation, and precipitation. Both the Piper diagram and HFE-D provide insights into the hydrochemical types of groundwater and highlight the roles of carbonate dissolution and recharge processes in shaping the groundwater composition. HFE-D offers a detailed view of chemical evolution through the interpretation of cation and anion percentage changes, making it suitable for identifying seawater intrusion and freshening stages [46–49]. In the intrusion phase, seawater infiltrates into the aquifer freshwater, which leads to the release of $Ca^{2+}$ and the adsorption of $Na^+$ ions through a reverse exchange process (R1) [50]. In the freshening process, a saline aquifer is replenished with fresh water abundant in calcium bicarbonate (R2), which results in the direct ion exchange [50] and transformation of the aquifer from an intrusion state to a freshening state.

R1: $Na^+ + Ca (Mg)\text{-}[clay] \rightarrow Ca^{2+} (Mg^{2+}) + [clay]\text{-}Na$      (reverse cation exchange)

R2: $Ca^{2+} (Mg^{2+}) + Na\text{-}[clay] \rightarrow Na^+ + [clay]\text{-}Ca (Mg)$      (direct cation exchange)

The HFE-D categorizes four distinct heteropic facies: Na-Cl for seawater, Ca-HCO$_3$ for natural freshwater, Ca-Cl for salinized water with reverse cation exchange, and Na-HCO$_3$ for salinized water with direct cation exchange. In the intrusion and freshening domains, distinct sub-stages can be discerned by tracing the changes in salinity (%Cl).

The chloro-alkaline indices (CAI), introduced by Schoeller (1965) [51], are represented by Equations (5) and (6) (with all ions stated in meq/L).

$$CAI\text{-}I = \frac{Cl^- - (Na^+ + K^+)}{Cl^-} \tag{5}$$

$$CAI\text{-}II = \frac{Cl^- - (Na^+ + K^+)}{HCO_3^- + SO_4^{2-} + CO_3^{2-} + NO_3^-} \tag{6}$$

The positive CAI indicates a reverse cation exchange reaction, i.e., Na+ and K+ in the groundwater are exchanged with $Ca^{2+}$ and $Mg^{2+}$ in the rocks. Conversely, the negative CAI indicates a cation–anion exchange reaction, where $Ca^{2+}$ and $Mg^{2+}$ in the groundwater are exchanged with $Na^+$ and $K^+$ in the rocks, particularly occurring under a chloro-alkaline disequilibrium condition.

## 3. Results and Discussion

### 3.1. Groundwater Geochemical Characteristics

The chemical compositions of groundwater samples from shallow and tube wells in the Mannar district are shown in Table 1 and Table S1. The groundwater in the study area exhibited a pH range of 5.89 to 7.97. Shallow wells showed a mean pH of 6.97, indicating near-neutral conditions that are likely related to carbonate mineral dissolution. Tube wells had a mean pH of 6.48, suggesting slight acidity that is possibly caused by the long geological contact with minerals and organic matter in deeper aquifers. In the Mannar district, shallow wells exhibited an average electrical conductivity (EC) of 1548.23 µS/cm, while tube wells showed a higher value of 2336.24 µS/cm. This difference possibly suggests that seawater intrusion resulted in elevated conductivity levels in coastal groundwater [52]. TDS is a fundamental parameter of water quality, and groundwater could be categorized as fresh water with TDS < 1 g/L and brackish water with TDS between 1 g/L and 3 g/L. Groundwater from shallow wells and tube wells in the Mannar district had an average TDS of 773.79 mg/L and 1168.35 mg/L, indicating that it mainly belonged to freshwater and brackish water. The higher TH (total hardness) [53] in tube wells (389.81 mg/L) than TH in shallow wells (291.00 mg/L) was consistent with TDS levels. The average value of EC, TDS, and TH surpassed the limits set by the World Health Organization (WHO) and the Sri Lanka Standards (SLS) for portable drinking water (Table S1), possibly suggesting the presence of seawater intrusion, evaporation, and weathering processes in the Mannar district. The DOC of shallow well groundwater (6.64 mg C/L) was obviously higher

than that of tube well groundwater (3.93 mg C/L), possibly implying a limited input of exogenous dissolved organic matter [54].

**Table 1.** Statistical analysis of geochemical characteristics.

| Parameters | Unit | Shallow Well | | | Tube Well | | |
|---|---|---|---|---|---|---|---|
| | | Range | Mean $\pm$ SD | CV | Range | Mean $\pm$ SD | CV |
| pH | | 5.92–7.97 | 6.97 $\pm$ 0.55 | 0.08 | 5.89–7.15 | 6.48 $\pm$ 0.40 | 0.06 |
| Temperature | °C | 27.30–28.20 | 27.63 $\pm$ 0.18 | 0.01 | 27.50–28.10 | 27.68 $\pm$ 0.16 | 0.01 |
| EC | μS/cm | 384.00–5910.00 | 1548.23 $\pm$ 1249.07 | 0.81 | 847.00–5310.00 | 2336.24 $\pm$ 1360.39 | 0.58 |
| TDS | mg/L | 192.00–2950.00 | 773.79 $\pm$ 624.13 | 0.81 | 424.00–2650.00 | 1168.35 $\pm$ 680.17 | 0.58 |
| $Na^+$ | mg/L | 4.77–797.00 | 162.20 $\pm$ 178.90 | 1.10 | 57.62–482.60 | 234.12 $\pm$ 156.83 | 0.67 |
| $K^+$ | mg/L | 0.29–68.01 | 16.21 $\pm$ 16.95 | 1.05 | 0.95–39.71 | 13.02 $\pm$ 11.58 | 0.89 |
| $Mg^{2+}$ | mg/L | 4.85–111.55 | 31.20 $\pm$ 20.60 | 0.66 | 14.43–100.37 | 43.95 $\pm$ 21.98 | 0.50 |
| $Ca^{2+}$ | mg/L | 17.54–187.83 | 65.09 $\pm$ 33.81 | 0.52 | 27.57–215.23 | 83.63 $\pm$ 48.07 | 0.57 |
| $HCO_3^-$ | mg/L | 190.32–980.88 | 405.48 $\pm$ 160.65 | 0.40 | 190.32–790.56 | 492.59 $\pm$ 121.01 | 0.25 |
| $F^-$ | mg/L | 0.14–2.48 | 0.64 $\pm$ 0.44 | 0.68 | 0.29–1.73 | 0.79 $\pm$ 0.43 | 0.54 |
| $Cl^-$ | mg/L | 7.38–1530.96 | 301.77 $\pm$ 368.19 | 1.22 | 35.80–1494.38 | 505.24 $\pm$ 430.86 | 0.85 |
| $Br^-$ | mg/L | 0–5.16 | 0.93 $\pm$ 1.11 | 1.20 | 0.15–5.50 | 1.75 $\pm$ 1.57 | 0.90 |
| $NO_3^-$ | mg/L | 0–85.60 | 4.92 $\pm$ 14.23 | 2.89 | 0–61.02 | 10.56 $\pm$ 16.64 | 1.58 |
| $SO_4^{2-}$ | mg/L | 0.99–906.70 | 70.91 $\pm$ 148.15 | 2.09 | 15.74–200.20 | 78.43 $\pm$ 61.52 | 0.78 |
| TH | mg/L | 84.81–928.36 | 291.00 $\pm$ 155.46 | 0.53 | 128.27–950.75 | 389.81 $\pm$ 192.97 | 0.50 |
| DOC | mg/L | 0.59–15.16 | 6.64 $\pm$ 3.19 | 0.48 | 1.26–9.15 | 3.93 $\pm$ 2.55 | 0.65 |
| WQI | | 29.01–306.54 | 100.12 $\pm$ 61.89 | 0.62 | 69.98–278.38 | 127.03 $\pm$ 61.91 | 0.49 |
| SMI | | 0.05–7.91 | 1.67 $\pm$ 1.95 | 1.17 | 0.51–6.82 | 2.60 $\pm$ 2.02 | 0.78 |

Notes: EC—electrical conductivity; TDS—total dissolved solids; TH—total hardness; DOC—dissolved organic carbon; WQI—water quality index; SMI—seawater mixing index; SD—standard deviation; CV—coefficient of variation.

The mean concentrations of $HCO_3^-$, $Cl^-$, $Na^+$, $SO_4^{2-}$, $Ca^{2+}$, $Mg^{2+}$, $K^+$, $NO_3^-$, $Br^-$, and $F^-$ ions in the shallow wells were 405.48, 301.77, 162.20, 70.91, 65.09, 31.20, 16.21, 4.92, 0.93, and 0.64 mg/L and 492.59, 505.24, 234.12, 78.43, 83.63, 43.95, 13.02, 10.56, 1.75, and 0.79 mg/L in the tube wells, respectively. This indicated the much higher content of major ions in tube well groundwater. It should be noted that the average concentrations of $Na^+$, $Cl^-$, $Ca^{2+}$, and $HCO_3^-$ surpassed the limits set by WHO and SLS for portable drinking water (Table S1). Based on the spatial distribution maps of geochemical characteristics (Figure 2), except for $HCO_3^-$, $F^-$, and DOC, most geochemical features, including other ions, EC, and TDS, had a similar distribution pattern. This manifested a low level of these features in most island areas and the central mainland and a high level in the north and south mainland regions. The overall level of EC, TDS, and major ions was higher in the south mainland regions than that in the north regions. TDS, $Na^+$, $Cl^-$, $Br^-$, $F^-$, and $HCO_3^-$ had a high level in the south and southeast mainland regions, while $NO_3^-$ and $F^-$ showed a high level in the east surface regions, which were closed to the Anuradhapura district [55]. This research by Hu et al. (2013) [55] indicated that the presence of $NO_3^-$ in the Anuradhapura district is primarily anthropogenic, e.g., from fertilizers [8]. The lower pH in tube wells, the higher EC and TDS levels in both shallow and tube wells, and the elevated concentrations of major ions such as $Cl^-$, $Na^+$, and $SO_4^{2-}$ in tube wells suggested the introduction of marine salts into the groundwater due to seawater intrusion in the Mannar district. Additionally, the higher coefficient of variation for specific ions in shallow

wells suggests greater variability, potentially indicating the impact of localized seawater intrusion points.

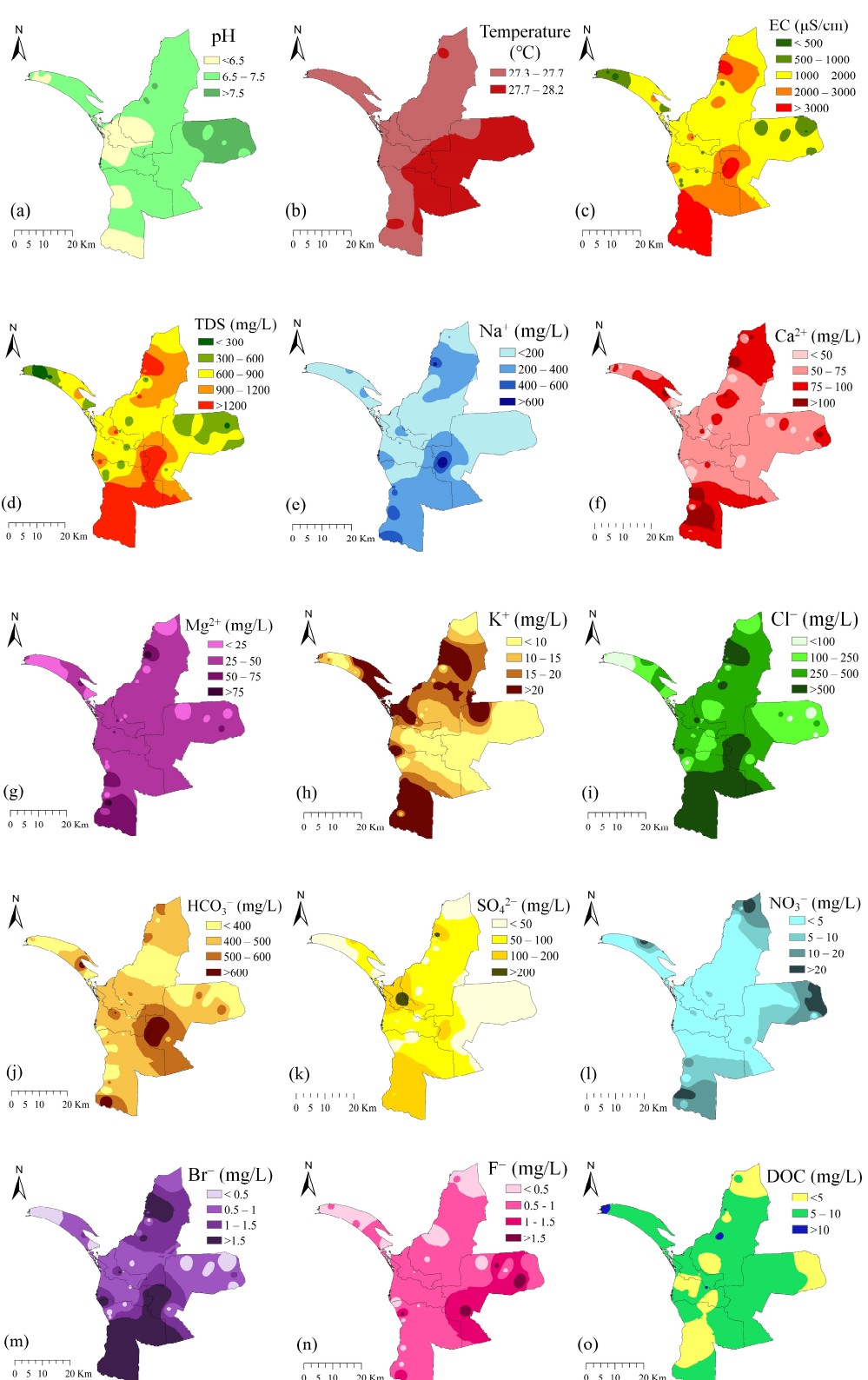

**Figure 2.** Distribution map of groundwater geochemical compositions. (**a**) pH, (**b**) temperature, (**c**) EC, (**d**) TDS, (**e**) Na$^+$, (**f**) Ca$^{2+}$, (**g**) Mg$^{2+}$, (**h**) K$^+$, (**i**) Cl$^-$, (**j**) HCO$_3^-$, (**k**) SO$_4^{2-}$, (**l**) NO$_3^-$, (**m**) Br$^-$, (**n**) F$^-$, and (**o**) DOC.

### 3.2. WQI and SMI

Based on the WQI data of groundwater in shallow and tube wells, 69.23% of shallow well groundwater and 52.94% of tube well groundwater in the Mannar area had excellent and good water quality (WQI < 100). This suggests that a portion of the groundwater still had poor and even worse water quality, which needed further treatment before drinking. The mean WQI of shallow and tube well groundwater was 100.12 and 127.03, respectively (Table 1), indicating a better water quality of groundwater from shallow wells than that from tube wells. The spatial distribution of WQI reveals that most islands, as well as the western and eastern mainland areas, exhibited higher-quality groundwater compared to other regions in the Mannar district (Figure 3a). The WQI had a similar distribution pattern with EC, TDS, $Na^+$, $F^-$, and $Cl^-$, indicating their great role in groundwater quality in the Mannar region.

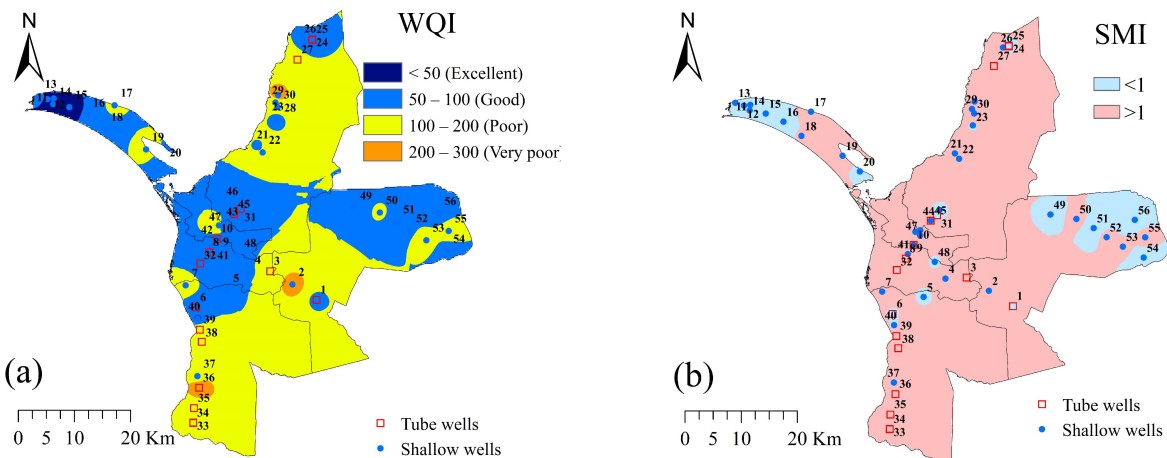

**Figure 3.** Distribution map of groundwater quality (**a**) and SMI (**b**) in the Mannar district.

The regional threshold value of SMI was determined to be 104 mg/L for $Na^+$, 27 mg/L for $Mg^{2+}$, 170 mg/L for $Cl^-$, and 45 mg/L for $SO_4^{2-}$, respectively. The SMI of groundwater ranged from 0.05 to 7.91, with an average value of 1.67 in shallow wells, and 0.51 to 6.82, with an average value of 2.60 in tube wells. Around 51.28% and 70.58% of groundwater samples in shallow wells and tube wells had an SMI of >1. These results indicate that many groundwaters might have undergone a certain degree of seawater intrusion, with tube well groundwater experiencing a heavier seawater intrusion than shallow well groundwater (Figure S2). The spatial distribution of SMI (Figure 3b) suggests that most regions besides the east mainland region and west island region were influenced by seawater mixing. Despite that, there was no direct relationship between WQI and SMI.

### 3.3. Groundwater Evolution Process and Mechanism

A Gibbs diagram (Figure 4) showed that $c(Na^+)/c(Na^+ + Ca^{2+})$ and $c(Cl^-)/c(Cl^- + HCO_3^-)$ of shallow groundwater were 0.06~0.95 (average 0.60) and 0.03~0.83 (average 0.37) and 0.34~0.90 (average 0.70) and 0.08~0.80 (average 0.47) of tube well groundwater, respectively. All the groundwater samples were distributed in the rock weathering-evaporative crystallization area, indicating the predominant influence of rock weathering and/or evaporation processes on the geochemical composition of groundwater. A small number of groundwater samples fell outside the boundaries of the Gibbs plots, and this might have arisen from anthropogenic activities or intense cation exchange [56]. The overall TDS and $c(Na^+)/c(Na^+ + Ca^{2+})$ of tube well groundwater were slightly higher than those of shallow well groundwater, possibly suggesting the susceptibility of seawater intrusion to tube well groundwater. In contrast, shallow well groundwater salt mainly originated from mineral weathering and was also affected by evaporation crystallization [57]. Unlike the rock weathering and/or evaporation processes, the precipitation had little effect on TDS, ($c(Na^+)/c(Na^+ + Ca^{2+})$), and $c(Cl^-)/c(Cl^- + HCO_3^-)$, which could be attributed to the

semi-arid climate during the dry season. End-member diagrams are used to distinguish the water interaction type with carbonate, silicates, and evaporite rocks [58]. Figure 4c,d showed that most groundwater in the Mannar area was located in and around the silicate dominance, indicating that silicate weathering (R3) was the predominant source of solutes for groundwater in the area, and evaporite and carbonate dissolution processes played a minor role in the geochemical compositions of groundwater in the Mannar district.

$$R3: 2NaAlSi_3O_8 + 2CO_2 + 11H_2O \rightarrow Al_2Si_2O_5(OH)_4 + 4H_4SiO_4 + 2Na^+ + 2HCO_3^-$$

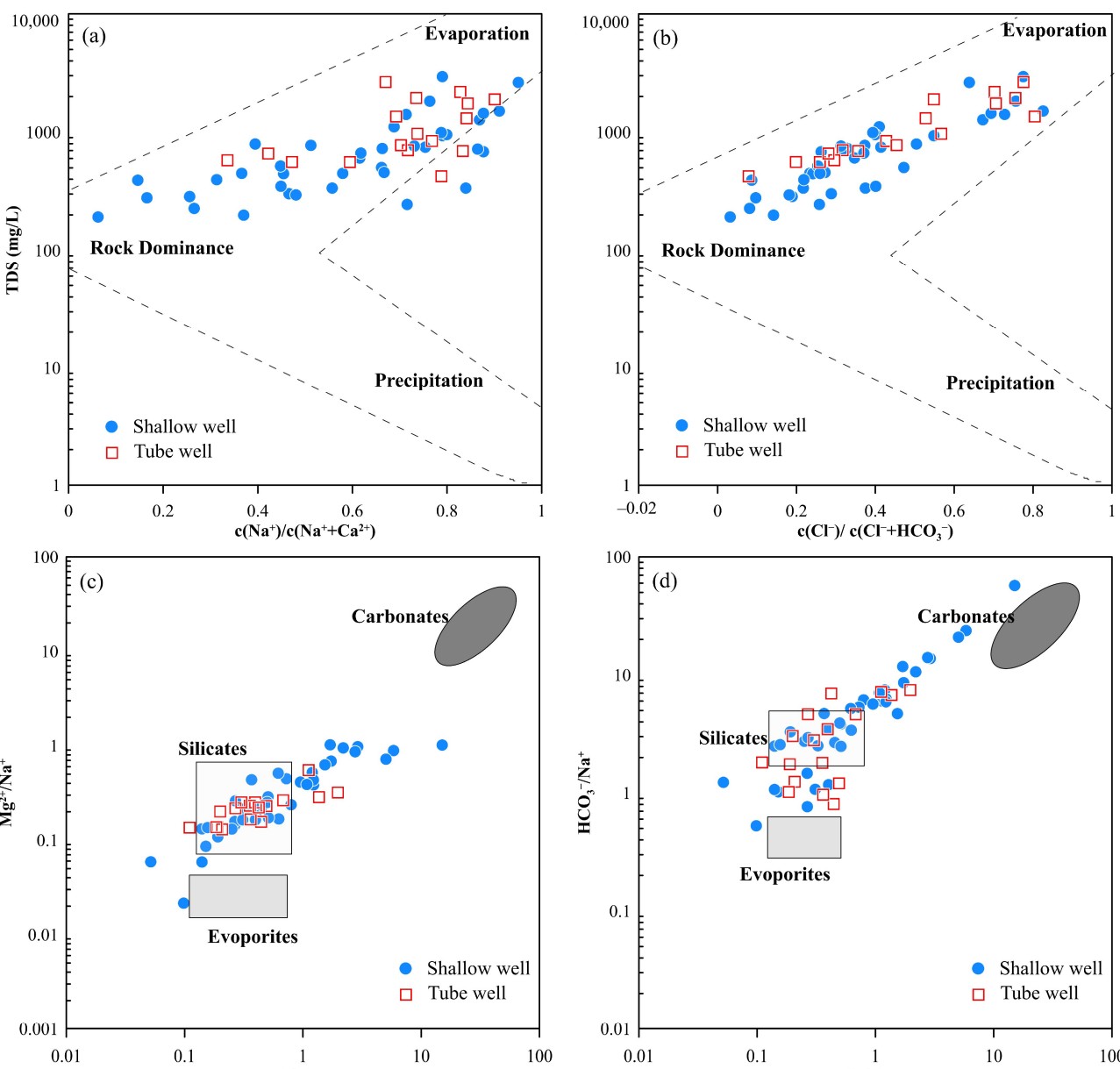

**Figure 4.** Gibbs (**a**,**b**) and end-member diagrams (**c**,**d**) of groundwater in the Mannar district.

### 3.4. Hydrochemical Types in Groundwater

A Piper diagram (Figure 5a) showed that Cl-Na- and HCO$_3$-Ca-type water facies were the main two hydrochemical types in the study area. Specifically, the HCO$_3$-Ca type, Cl-Na type, and mixed type accounted for 41.02%, 38.46%, and 20.51% for shallow well groundwater and 23.53%, 58.82%, and 17.64% for tube well groundwater. The Cl-Na type

reflects the dominance of Na$^+$ and Cl$^-$ ions and is indicative of seawater influence or arid conditions.

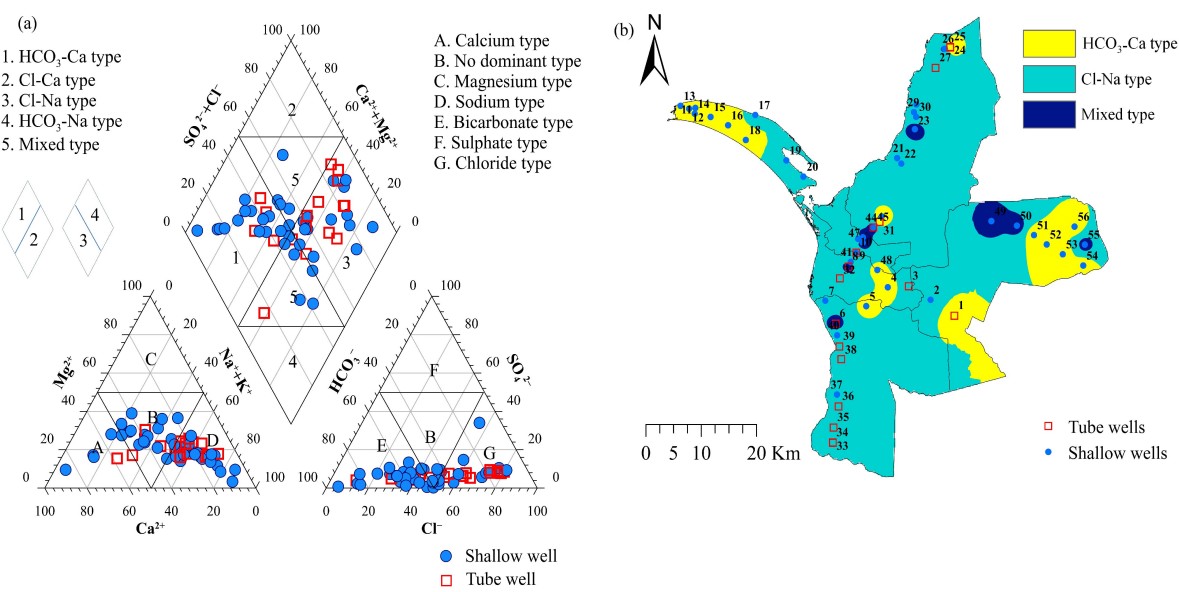

**Figure 5.** Piper diagram (**a**) and spatial distribution (**b**) of groundwater types.

About 44.64% of shallow groundwater in the Mannar district was Cl-Na type (Figure 6), which was more evident for tube well groundwater (58.82%). The Cl-Na-type groundwater was mainly distributed in the northern and southern inland areas, and the majority had a high concentration of Cl$^-$ and Na$^+$. The left deviation of $c(Cl^-)/c(Na^+)$ to 1 mainly occurred in the area with high content of Cl$^-$ and Na$^+$ (Figure S4a). This indicates that the concentrations of Cl$^-$ and Na$^+$ were higher in the groundwater suffering from seawater intrusion than those from halite dissolution. Around 46.15% of shallow well samples and 71.43% of tube well samples had $c(Cl^-)/c(Na^+)$ values lower than the seawater value (1.17), also suggesting the potential mixing of seawater with freshwater [19]. In combination with Figure 2, the areas suffering from seawater intrusion are mainly located in the northern and southern mainland areas. A positive value of both CAI-I and CAI-II was also observed for these groundwaters (Figure S4d), indicating the occurrence of the reverse cation exchange between Na$^+$ and K$^+$ in groundwater with Ca$^{2+}$/Mg$^{2+}$ from aquifer materials [59]. These results implied that the high content of Cl$^-$ and Na$^+$ in groundwater might be from intrusive seawater and favorable for reverse cation exchange [10], leading to a decrease in Na$^+$/K$^+$ and an increase in Ca$^{2+}$/Mg$^{2+}$.

HCO$_3$-Ca-type water facies are mainly distributed in the western island and the east and southeast mainland areas (Figure 5b). It has been reported that Ca$^{2+}$, Mg$^{2+}$, and HCO$_3$$^-$ were mainly from the dissolution of carbonates from carbonate-rich rocks such as limestone (Reactions R4 and R5, Figure 4) [60,61], which were also abundant in the above areas. The sampling points belonging to the HCO$_3$-Ca type usually had a low content of Cl$^-$ and Na$^+$ (Figure S4a), indicting the little effect of seawater intrusion on these areas. Most HCO$_3$-Ca-type groundwater had $c(HCO_3^-)/c(Ca^{2+})$ in the range of 1 to 2 and $c(HCO_3^-)/c(Ca^{2+} + Mg^{2+})$ close to 1 (Figure S4b,c), implying that these ions mainly come from the dissolution of silicate, gypsum, and limestone.

$$R4: CaCO_3 + CO_2 + H_2O \rightarrow Ca^{2+} + 2HCO_3^-$$

$$R5: Ca.Mg(CO_3)_2 + 2CO_2 + 2H_2O \rightarrow Ca^{2+} + Mg^{2+} + 4HCO_3^-$$

Approximately 17.86% of the groundwater samples were mixed-type water facies and were mainly distributed in the middle of the inland area and the east part of the district.

Groundwater chemistry was jointly affected by the dissolution of mineral rocks, seawater intrusion, and ion exchange processes [62].

| Sub stage | Anion facies | Ion percentage |
| --- | --- | --- |
| Freshwater | -HCO₃ | % HCO₃ > 50% <br> %Ca > 66.6% |
| f4 | -HCO₃ | % HCO₃ > 50% |
| f3 | -MixHCO₃ | 50% ≥ % HCO₃ > 33.3% |
| f2 | -MixCl | 33.3% < %Cl ≤ 50% |
| f1 | -Cl | 50% < %Cl ≤ 66.6% |
| i1 | -MixHCO₃ | 50% ≥ % HCO₃ > 33.3% |
| i2 | -MixCl | 33.3% < %Cl ≤ 50% |
| i3 | -Cl | 50% < %Cl ≤ 66.6% |
| i4 | -Cl | %Cl < 66.6% |
| Saltwater | -Cl | %Cl < 66.6% <br> %Na > 50% |

**Figure 6.** HFE-D of groundwater in the Mannar district. (**a**) Main hydrochemical facies and sub-stages for intrusion and freshening phases (modified from Giménez-forcada, E., 2019) [49]. (**b**) Distribution map of facies.

### 3.5. Chemical Evolution of Groundwater

The more detailed distribution of hydrochemical facies for shallow and tube well groundwater was presented using the HFE-Diagram (Figure 6a). As shown in Figure 6a, the freshening sub-stage consisting of f1, f2, f3, f4, and FW is located within the FwI phase, marked by the curved arrow on the upper left. Conversely, the intrusion phase is depicted in the opposite sector of the diagram, with the SwI sub-stages (i1, i2, i3, i4, and SW). For the shallow well groundwater, nine sub-stages of hydrochemical facies were identified (Table S3), with freshening and intrusion phases accounting for 74.36% and 25.64%, respectively, while for the tube well groundwater, seven sub-stages of hydrochemical facies were identified, with freshening and intrusion phases accounting for 64.71% and 35.29%, respectively.

Generally, the intrusion phase mainly occurred on the southern and northern mainlands of the Manna district (Figure 6b). The color from light to dark indicated the evolutionary processes and impacting degree of seawater intrusion, i.e., started from inland distal intrusion facies, MixNaMixCl, MixCaMixCl, gradually evolved to sub-stages of MixCaCl and MixNaCl, and finally became Na-Cl, which represented the final mixture and tendency of the groundwater components similar to the seawater (Figure 6a). The area belonging to the intrusion phase was involved in the seawater intrusion and reverse cation exchange processes, which contributed to the increasing salinity of groundwater. The reverse cation exchange process led to the conversion of Na-Cl groundwater into Ca-Cl groundwater as seawater intrusion progressed with an adequate supply of $Ca^{2+}$ in sandy aquifer minerals. Most regions in the Mannar district, including the island section and central and eastern mainland areas, were associated with the freshening phase (Figure 6b), further indicating the major contribution of $Na^+$ and $Cl^-$ from the halite dissolution and of $HCO_3^-$ from the silicate dissolution. As a mineral mainly composed of sodium chloride, halite usually presents in geological formations such as salt domes, salt pans, or salt marshes, and its dissolution process, thus, has an important impact on the concentration of $Na^+$ and $Cl^-$. In the areas with relatively higher $HCO_3^-$ and $Ca^{2+}$, the direct exchange processes (R2) also made a contribution to the $Na^+$ and $K^+$; however, this process might only play a minor role due to the small portion of HCO₃-Ca-type groundwater in the Mannar district.

## 4. Conclusions

In summary, this study explored the geochemical compositions and evolution processes of groundwater in the Mannar district of Sri Lanka. $HCO_3^-$, $Cl^-$, $Na^+$, $SO_4^{2-}$, $Ca^{2+}$, $Mg^{2+}$, $K^+$, $NO_3^-$, $Br^-$, and $F^-$ were the main ions in the groundwater, and their concentrations were higher in the groundwater of tube wells than shallow wells. Approximately 64.28% of the groundwater samples had good quality based on a WQI value below 100. Seawater intrusion had a significant impact on 51.28% of shallow well groundwater and 70.58% of tube well groundwater. It should be noted that groundwater suffering from seawater intrusion does not mean poor water quality due to the calculation difference between SMI and WQI. The geochemical composition of groundwater in the Mannar district was predominantly influenced by the processes of rock weathering and/or evaporation. The dissolution of halite and silicate mainly formed the geochemical types of groundwater, with $HCO_3$-Ca type, Cl-Na type, and mixed type accounting for 41.02%, 38.46%, and 20.51% of shallow well groundwater and 23.53%, 58.82%, and 17.64% of tube well groundwater. Seawater intrusion primarily affected the northern and southern mainland regions, leading to elevated levels of $Cl^-$, $Na^+$, and other main ions, which, in turn, triggered a reverse cation exchange process. Nine and seven sub-stages of hydrochemical facies, as well as the contribution of freshening and seawater intrusion phases, were revealed for shallow and tube well groundwater. $HCO_3$-Ca and Cl-Na facies accounted for 74.36% and 64.71% of freshening phases, as well as 25.64% and 35.29% of intrusion phases for shallow well and tube well groundwaters, respectively. These findings emphasize the significant impact of seawater intrusion on the geochemical compositions and evolution processes of groundwater in a typically tropical region and will contribute to the effective management and treatment of groundwater resources in a sustainable manner, such as alleviating seawater intrusion, routinely monitoring the groundwater quality, and implementing targeted desalination techniques.

**Supplementary Materials:** The following supporting information can be downloaded at https://www.mdpi.com/article/10.3390/w16081137/s1, Figure S1: East-West geological schematic cross sections of (a) deep confined aquifer basin system of Mannar mainland and (b) sandy aquifer systems of Mannar Island; Figure S2: Cumulative probability (%) for calculating the threshold value of critical parameters ($Na^+$, $Mg^{2+}$, $Cl^-$, $SO_4^{2-}$); Figure S3: SMI values of groundwater in the Mannar district; Figure S4: Hydrochemical relationships between ions. Orange, green, and purple colors represent $HCO_3$-Ca type, Cl-Na type, and mixed type, respectively. Solid and hollow points correspond to shallow well and tube well groundwater, respectively. (a) $Cl^-$ vs. $Na^+$. (b) $HCO_3^-$ vs. $Ca^{2+}$. (c) $HCO_3^-$ vs. ($Ca^{2+}$ + $Mg^{2+}$). (d) Chloro-alkaline index diagram CAI II vs. CAI I; Table S1: Statistics of groundwater geochemical compositions; Table S2: Relative weight of chemical parameters with threshold values; Table S3: Sub-stages (in %) in freshening and intrusion phases.

**Author Contributions:** All authors contributed to the study's conception and design. S.A. and Y.W. (Yunwen Wang) drafted the manuscript, performed data analysis, and prepared figures and maps. S.I. and I.Y. assisted with the sample collection S.K.W. and J.L. conducted the investigation, developed the methodology, and carried out the validation. Z.H. and Y.W. (Yuansong Wei) were responsible for project administration, supervision, as well as the writing, review, and editing of the manuscript. All authors have read and agreed to the published version of the manuscript.

**Funding:** The authors would like to acknowledge the financial support from the National Natural Science Foundation of China (22176199), the Joint Research Program of National Natural Science Foundation of China and National Science Foundation of Sri Lanka (NSFC-NSF SL) (21861142020), the Programme of the Comprehensive Studies on Sri Lanka (059GJHZ2023104MI), the Research Center for Eco-Environmental Science (RCEES-TDZ-2021-14), the Alliance of International Science Organizations Collaborative Research Program (ANSO-CR-KP-2020-05), the Program of China–Sri Lanka Joint Research and Demonstration Center for Water Technology, the China–Sri Lanka Joint Center for Education and Research, the Chinese Academy of Sciences (CAS), and the Alliance of International Science Organizations (ANSO) Scholarship for Young Talents (MSc) (series No. 2020-140).

**Data Availability Statement:** Data is contained within the article or Supplementary Materials.

**Acknowledgments:** We would like to thank the staffs and people of China-Sri Lanka Joint Research and Demonstration Center for Water Technology who supported the sample collection. Anonymous reviewers were greatly appreciated for their constructive comments, which helped in improving the quality of the article.

**Conflicts of Interest:** The authors declare no conflict of interest.

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
