# Peer review of "Geochemical Assessment of the Evolution of Groundwater under the Impact of Seawater Intrusion in the Mannar District of Sri Lanka"

_water, doi:10.3390/w16081137_

Round 1

Reviewer 1 Report

Comments and Suggestions for Authors

The article "Geochemical assessment of the Evolution of Groundwater under the Impact of Seawater Intrusion in the Mannar District of Sri Lanka", written by authors Samadhi Athauda, Yunwen Wang, Zhineng Hao, Suresh Indika, Isuru Yapabandara, Sujithra K. 5 Weragoda, Jingfu Liu and Yuansong Wei, representing scientific institutions: Research Center for Eco-Environmental Sciences, Chinese Academy of Sciences, Beijing (China), University of Chinese Academy of Sciences, Beijing (China), China-Sri Lanka Joint Research and Demonstration Center for Water Technology , Ministry of Water Supply (Sri Lanka), National Water Supply and Drainage Board, Katugastota (Sri Lanka), Institute of Environment and Health, Jianghan University, Wuhan (China), is methodical and a case study.

This study analyzed hydrogeochemical compositions, water quality, and evolution processes of groundwater in Mannar district, Sri Lanka, during the dry season. As a result of the conducted research, it was found that the geochemical compositions of groundwater in Mannar district were largely affected by rock weathering and/or evaporation processes. Cl-Na and HCO3-Ca facies were the main hydrochemical types, and the corresponding ions were mainly from silicate and halite dissolution. Reverse cation exchange process mainly occurred in the seawater intrusion areas. The study highlights the impacts of seawater intrusion on hydrogeochemical compositions and the evolution processes in Mannar region groundwater,

Research results, discussion, using modern research techniques, and conclusions are logical at a good scientific level.

The following are critical notes:

 Abstract

1/ "The study highlights the impacts of seawater intrusion on hydro geochemical compositions and the evolution processes in Mannar region groundwater, which will aid in the sustainable groundwater resource management and treatment.." - the authors write in many articles about the possibility of applying the obtained research results in practice – decisions and actions. These are very general statements. Or write down specific decisions and actions that may be taken or these types of statements removed.

1. Introduction

1/ "The northern and northwestern regions of Sri Lanka features with permeable coastal aquifers that are composed of a Miocene limestone belt and unconsolidated materials." - what hydrogeological parameters characterize the aquifer?

2/ Why was May chosen for the study, apart from the fact that it is the dry season? Is this when the intensity of seawater intrusion into aquifers is greatest? It was possible to perform several tests at different times during the dry season to monitor the dynamics of hydrogeochemical processes. At the same time, it should be noted that seawater intrusions are a seasonal phenomenon. Is there intensive groundwater exploitation with wells in the research area that favors seawater intrusion?

 2. Experimental section

1/ "...an average annual rainfall of 975 mm/year and evapotranspiration of 2135 mm/year." Please present the annual general water balance, as the data presented are inconsistent. What components are on the recharge side in the general water balance?

2/ Briefly describe the geological and hydrogeological characteristics of the research area, including a figure showing hydrological conditions. Are there two aquifers in the study area - a shallower and a deeper one? The hydrogeological cross-section could show the phenomenon of seawater intrusion into aquifers. Without this brief natural characterization of the research area, the article is only "technical" and focuses solely on methodological aspects.

 2.2. Sample collections and measurements

1/ Write in which laboratory the hydrochemical tests were performed

4. Conclusions

1/ “Approximately 64.28% of the groundwater samples had good quality based on the WQI value below 100. Seawater intrusion significantly impacted 51.28% of the shallow well groundwater and 70.58% of the tube well groundwater.” If seawater intrusion significantly impacted more than 50% of the wells, how did 64.28% of the groundwater samples have good quality?

2/ "...The dissolution of halite and silicate mainly formed the geochemical types of groundwater." - in what geological formation does halite occur?

3/ "These findings emphasize the significant impact of seawater intrusion on the geochemical compositions and evolution processes of groundwater in a typically tropical region and will contribute to the effective management and treatment of groundwater resources in a sustainable manner." – what would the effective management and treatment of groundwater resources in a sustainable manner involve?

 References

Almost all references in References are cited in the text, except reference 34.

The article is of good scientific quality. It requires supplementation in the natural and environmental aspects. It presents a certain representativeness of the research results for a specific natural, geological, hydrological environment and climatic conditions (?). At the same time, it refers to published and already used research methods. The authors should clearly indicate what the scientific originality of the article is. The article should find
a wide range of readers involved in hydrological research environmental threats, including seawater intrusion threats, using modern research methods in hydrogeochemistry.

Reviewer 2 Report

Comments and Suggestions for Authors

I have reviewed the manuscript “

 Geochemical assessment of the Evolution of Groundwater under the Impact of Seawater Intrusion in the Mannar District of 3 Sri Lanka, and my comments are as follows;

1. Page No. 3, Please change the map and it should be clearly legible.

2. Page No. 6. In the data table, please follow the pattern, pH, EC, TDS, cations, Anions, and TH. There should not arbitrarily presented the ions.

3. The major ions should be compared with drinking water standards of WHO.

4. The fractions after the number 20 should be rounded.

5. In the page No. 8, section 3.3, there should be a cross plot of Na vs. Cl, and the ratio should be used for the seawater percentage. Can refer a paper,

Mohanty, A. K., and Gurunadha Rao, (2019). Hydrogeochemical, seawater intrusion and oxygen isotope studies on a coastal region in the Puri District of Odisha, India. CATENA, 172, 558 -571.

After the necessary modifications, the manuscript can be accepted for the publication.

Comments on the Quality of English Language

Quality of english is to be improved. 

Round 2

Reviewer 2 Report

Comments and Suggestions for Authors

The authors have incorporated all the comments and suggestions. The manuscript can be accepted.